# Ibrutinib reverses IL-6-induced osimertinib resistance through inhibition of Laminin α5/FAK signaling

Li Li [1,4✉], Zhujun Li[1,4], Conghua Lu [1,4], Jianghua Li[1], Kejun Zhang[2], Caiyu Lin[1], Xiaolin Tang[1], Zhulin Liu[1], Yimin Zhang[1], Rui Han[1], Yubo Wang[1], Mingxia Feng[1], Yuan Zhuang [3], Chen Hu [1✉] & Yong He [1✉]

Osimertinib, a 3rd generation epidermal growth factor receptor tyrosine kinase inhibitor (EGFR-TKI), is the first-line standard-of-care for EGFR-mutant non-small cell lung cancer (NSCLC) patients, while acquired drug resistance will inevitably occur. Interleukin-6 (IL-6) is a keystone cytokine in inflammation and cancer, while its role in osimertinib efficacy was unknown. Here we show that clinically, plasma IL-6 level predicts osimertinib efficacy in EGFR mutant NSCLC patients. Highly increased IL-6 levels are found in patients with acquired resistance to osimertinib. Addition of IL-6 or exogenous overexpression of IL-6 directly induces osimertinib resistance. Proteomics reveals LAMA5 (Laminin α5) and PTK2, protein tyrosine kinase 2, also called focal adhesion kinase (FAK), are activated in osimertinib-resistant cells, and siRNA knockdown of *LAMA5* or *PTK2* reverses IL-6-mediated osimertinib resistance. Next, using a large-scale compound screening, we identify ibrutinib as a potent inhibitor of IL-6 and Laminin α5/FAK signaling, which shows synergy with osimertinib in osimertinib-resistant cells with high IL-6 levels, but not in those with low IL-6 levels. In vivo, this combination inhibits tumor growth of xenografts bearing osimertinib-resistant tumors. Taken together, we conclude that Laminin α5/FAK signaling is responsible for IL-6-induced osimertinib resistance, which could be reversed by combination of ibrutinib and osimertinib.

[1] Department of Respiratory Medicine, Daping Hospital, Third Military Medical University (Army Medical University), Chongqing 400042, China. [2] Department of Clinical Laboratory, Daping Hospital, Third Military Medical University (Army Medical University), Chongqing 400042, China. [3] National Engineering Research Center of Immunological Products, Department of Microbiology and Biochemical Pharmacy, College of Pharmacy and Laboratory Medicine, Third Military Medical University (Army Medical University), Chongqing 400038, China. [4] These authors contributed equally: Li Li, Zhujun Li, Conghua Lu. ✉email: dpyyhxlili@tmmu.edu.cn; huchen89@tmmu.edu.cn; heyong@tmmu.edu.cn

Epidermal growth factor receptor (EGFR) mutations are the most common actionable driver mutation in metastatic non-small-cell lung cancer (NSCLC). Despite the vast majority of EGFR mutant lung cancer patients treated with EGFR tyrosine kinase inhibitors (TKIs) derive clinical benefit, acquired drug resistance is inevitable. More recently, osimertinib, a 3rd generation EGFR-TKI, has been established as first-line therapy based on improved outcomes compared to 1st and 2nd generation TKIs[1]. Therefore, there is a pressing need to understand acquired mechanisms of resistance to osimertinib.

The resistance mechanisms of 1st and 2nd generation TKIs have been extensively studied, with EGFR T790M as the most common mechanism. However, acquired resistance mechanisms to osimertinib have not been fully elucidated[2]. A series of clinical and preclinical studies have revealed several potential resistance mechanisms, including EGFR-dependent mechanisms such as resistance mutations in EGFR (e.g., C797S or L718Q), and EGFR-independent mechanisms including activation of alternate RTKs (e.g., MET, FGFR, and IGF1R), PIK3CA mutations, RAS/RAF mutations, and Exon 16-Skipping HER2, as well as non-oncogene-dependent mechanisms, including epithelial-mesenchymal transition (EMT), acquisition of stem-like properties, and metabolic rewiring[3–6]. However, effective therapeutic strategies targeting these mechanisms are still lacking.

In recent years, interleukin-6 (IL-6) has been indicated as a potential resistance mechanism to EGFR-TKIs. IL-6 represents a keystone cytokine in infection, cancer, and inflammation, which can be produced by multiple cell types in the tumor microenvironment, leading to JAK/STAT3 signaling activation in both tumor cells and tumor-infiltrating immune cells, and promoted the proliferation, survival, invasiveness, and metastasis of tumor cells, while strongly suppressing the anti-tumor immune response[7,8]. Previously, we and others have reported that IL-6 could drive resistance to 1st generation EGFR-TKI through activation of STAT3 and promoting EMT[9,10]. In EGFR-mutant NSCLC treated with first-line 1st generation EGFR-TKIs, a high baseline level of serum IL-6 was associated with shorter progression-free survival (PFS)[11,12]. However, it is unknown whether IL-6 may influence osimertinib efficacy or be associated with osimertinib resistance.

Herein, the current study aimed to identify whether IL-6 was correlated with osimertinib efficacy and found that Laminin α5/FAK signaling activation mediates IL-6-induced osimertinib resistance, which could be overcame by ibrutinib in combination with osimertinib.

## Results

### Preclinical and clinical evidence that IL-6 was strongly associated with osimertinib resistance.
To screen potential resistance mechanism to osimertinib, a high-throughput transcriptome sequencing (RNA-Seq) was performed in 3 paired cell strains sensitive/resistant to osimertinib. We cultured PC-9 cells, PC-9GR cells, and H1975 cells (all EGFR mutant) with increasing concentrations of osimertinib until resistant variants emerged (PC-9OR cells, PC-9GROR cells, and H1975OR cells). The coverage and overlap of DEGs were illustrated in the Venn diagram, and a total of 878 DEGs were commonly identified in all groups (Fig. 1a and Supplementary Data 1). Then, protein–protein interaction (PPI) network was drawn to explore how these DEGs interact with each other and it turns out that IL-6 was with the highest degree (Fig. 1a). Consistent with the RNA-Seq results, higher levels of IL-6 were found in culture medium as well as cell lysates of osimertinib-resistant cell lines by ELISA (Fig. 1b). These results suggest that IL-6 was associated with osimertinib resistance.

We then analyzed the association between pretreatment plasma IL-6 levels with EGFR-TKI efficacy. A total of 286 treatment-naïve EGFR mutant NSCLC patients with available pretreatment plasma IL-6 levels were screened. Of them, 25 were excluded, and the remaining patients were divided into two groups: 233 receiving gefitinib as 1st line therapy, and 28 receiving osimertinib as 1st line therapy (Supplementary Fig. 1a and Supplementary Table 1). In the 226 patients treated with gefitinib with response evaluation, higher IL-6 ($\geq$ 7 mg/L) was associated with a worse PFS compared to patients with lower IL-6 concentrations (Fig. 1c). Similarly, in those 70 patients receiving osimertinib (including 1st line and 2nd line treatments), higher pre-osimertinib IL-6 ($\geq$ 7 mg/L) was also associated with a worse PFS (Fig. 1d). These data suggest that baseline plasma IL-6 level predicts EGFR-TKI efficacy in EGFR mutant NSCLC patients.

We next studied IL-6 dynamics during TKI treatment. Paired plasma IL-6 levels at baseline and disease progression were available in a total of 126 patients treated with gefitinib and 39 patients treated with osimertinib. Highly increased IL-6 levels were found upon acquired resistance to either gefitinib or osimertinib (Fig. 1e). Moreover, a higher portion of patients experienced IL-6 increase upon osimertinib resistance compared with gefitinib resistance (82.1% vs 62.7%, $p = 0.024$) and IL-6 levels upon osimertinib resistance were also higher than gefitinib resistance ($p = 0.001$). We next assessed the overall survival (OS) of patients from resistance to gefitinib or osimertinib according to IL-6 dynamics. Upon gefitinib resistance, patients with highly elevated IL-6 levels (value $\geq$ 7 mg/L upon resistance and higher than that of baseline) had a shorter OS than those with decreased or slightly increased IL-6 levels (value < 7 mg/L upon resistance), as shown in Supplementary Fig. 1b. Upon osimertinib resistance, patients with highly elevated IL-6 levels had a non-significant shorter OS, possibly due to limited patient numbers (Supplementary Fig. 1c).

We then assessed the association of IL-6 level upon TKI resistance with currently known drug resistance mechanisms. A tissue or plasma-based next-generation sequencing (NGS) was performed to find potential resistance mechanisms in a total of 88 patients with resistance to gefitinib, and 26 patients with osimertinib resistance. Interestingly, in those with resistance to gefitinib, the resistance mechanisms were similar between those with high or low IL-6 levels (Fig. 1f). However, in those with resistance to osimertinib, a higher portion of patients with high IL-6 levels developed unknown mechanisms (Fig. 1g). Taken together, these data suggest IL-6 was strongly associated with osimertinib resistance clinically and pre-clinically.

We then asked whether IL-6 may directly induce osimertinib resistance. Addition of IL-6 into the culture medium decreased the osimertinib sensitivity of PC-9 cells, PC-9GR cells, H1975 cells, HCC827 cells, and H3255 cells (Fig. 1h and Supplementary Fig. 2a, b). Similarly, overexpression of exogenous IL-6 (IL-6-GFP PC-9 cells, IL-6-GFP PC-9GR cells, and IL-6-GFP HCC827 cells) also weakened osimertinib sensitivity (Fig. 1h and Supplementary Fig. 2a, d). Then, the Ki67 incorporation assay was applied to measure cell proliferation. In sensitive PC-9 cells, PC-9GR cells, HCC827 cells, and H3255 cells, osimertinib treatment decreased the percentage of Ki67-positive cells, while IL-6 addition abrogated this effect. Also, treatment of osimertinib showed little effect on the percentage of Ki67-positive cells in IL-6-GFP PC-9 cells and IL-6-GFP PC-9GR cells (Fig. 1i and Supplementary Fig. 2c–f). We then performed flow cytometry analysis of Annexin V and PI to detect apoptosis in PC-9 cells and H3255 cells, treated with osimertinib, IL-6, or both. As shown in Supplementary Fig. 3, osimertinib treatment induced apoptosis in both cell lines. However, in the presence of IL-6, osimertinib failed to induce apoptosis of these cells. Taken together, these results suggest that IL-6 can directly induce osimertinib resistance.

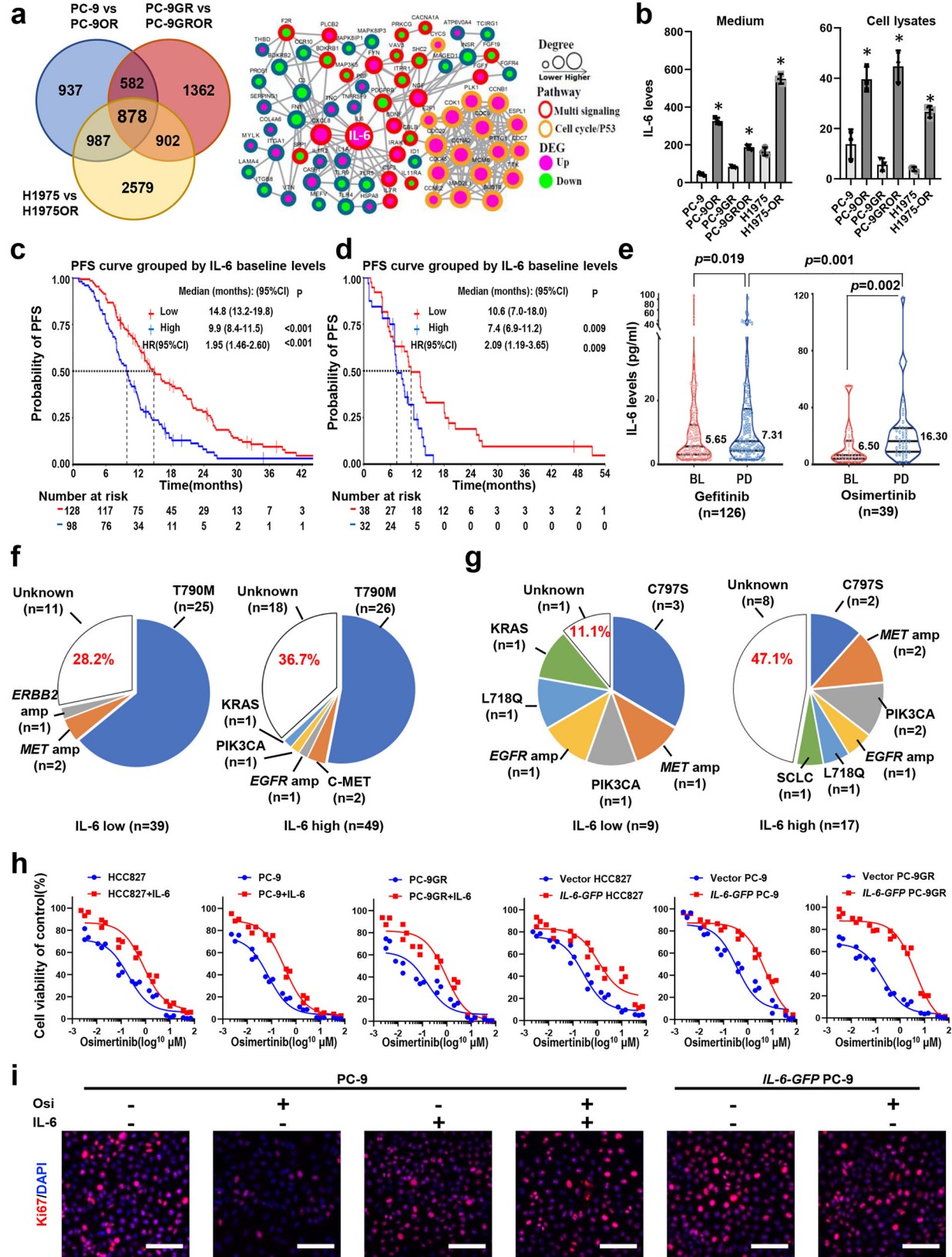

**Proteomics revealed that Laminin α5 was required for IL-6-induced osimertinib resistance.** We then asked whether inhibition of IL-6 signaling can overcome osimertinib resistance. We first studied whether tocilizumab, a neutralizing IL-6R antibody, could reverse IL-6-induced osimertinib resistance. In vitro, CCK-8 assay showed that tocilizumab failed to increase osimertinib sensitivity in

PC-9GROR cells (Supplementary Fig. 4a, b). Next, in mice bearing PC-9GROR xenografts, the combination of tocilizumab and osimertinib did not inhibit tumor growth (Supplementary Fig. 4e–h). Similarly, STAT3 inhibitor STATTIC also showed little effect on osimertinib sensitivity in osimertinib-resistant cells (Supplementary Fig. 4c, d). Previously, it was reported that tocilizumab could

**Fig. 1 IL-6 was closely associated with osimertinib resistance. a** High-throughput transcriptome sequencing in 3 paired cell strains sensitive/resistant to osimertinib, respectively, as indicated. Left: The Venn diagram of the commonly and exclusively differentially expressed genes (DEGs), and the overall overlapping site refers to the DEGs commonly found in all groups. Right: The protein–protein interaction network of 233 DEGs. Red and green nodes represent upregulated and downregulated genes after TKI resistance, respectively. The size of nodes refers to the degree of interaction with other genes. GR, gefitinib resistance; OR, osimertinib resistance. **b** ELISA assay of IL-6 levels in culture medium (pg/ml) or cell lysates (pg/100 μg protein) from paired osimertinib-sensitive and osimertinib-resistant cells, as indicated ($n = 3$ biologically independent experiments). Data are shown as mean ± SEM. *, $p < 0.01$ as compared to each parental cell strains. **c** and **d** Kaplan–Meier (KM) estimates of PFS in NSCLC patients treated with gefitinib, or osimertinib, respectively, according to high (greater than 7 mg/L) or low baseline IL-6 levels; **e** paired plasma IL-6 levels at baseline and disease progression of patients treated with gefitinib ($n = 126$) or osimertinib ($n = 39$). Lines indicate median and interquartile range of each group. P-value was calculated using a Mann–Whitney test. BL, baseline; PD, progression of disease. **f** gefitinib resistance mechanisms in low IL-6 group ($n = 39$) and high IL-6 group ($n = 49$), respectively. **g** Osimertinib resistance mechanisms in low IL-6 group ($n = 9$) and high IL-6 group ($n = 17$), respectively. **h** Cell viability CCK-8 assay for cells with different treatments as indicated. HCC827 cells, PC-9 cells, or PC-9GR cells were treated with IL-6 (20 ng/ml) together with increasing concentrations of osimertinib for 48 h. IL-6-GFP PC-9 cells, IL-6-GFP PC-9GR cells, and IL-6-GFP HCC827 cells were treated with indicated concentrations of osimertinib for 48 h. Data are shown as mean ± SEM ($n = 3$ biologically independent experiments). **i** Ki67 incorporation assay on PC-9 cells and IL-6-GFP PC-9 cells with different treatments as indicated. IL-6 (20 ng/ml) or osimertinib (1 μM) were added to the culture medium for 48 h. Cells were counterstained with DAPI. Scale bars: 100 μm.

overcome IL-6-induced resistance to erlotinib, a 1st generation EGFR-TKI[13]. Also, we and others have reported that inhibition of STAT3 could reverse drug resistance to 1st generation EGFR-TKIs[10,14]. These data suggest that there is a difference of mechanism between 1st/2nd generation EGFR-TKI and osimertinib. IL-6/IL-6R blockade or STAT3 inhibition alone was unable to reverse osimertinib resistance.

To explore the potential mediator of IL-6-induced osimertinib resistance, we next performed mass-spectrometry-based quantitative proteomics to quantify protein expression in PC-9GR cells and PC-9GROR cells (Fig. 2a and Supplementary Data 2). A total of 1137 differentially-regulated proteins (421 downregulated and 716 upregulated) were quantified and mapped onto a volcano plot (Fig. 2b). Four IL-6 downstream pathways, MAPK signaling pathway, JAK-STAT pathway, NF-kB pathway, and PI3K-AKT pathway were upregulated by KEGG pathway enrichment analysis and gene set enrichment analysis (Supplementary Fig. 5a, b), and most proteins in those pathways were upregulated, as shown in the volcano plot and the heatmaps (Fig. 2b, c). Of note, Laminin α5 (LAMA5) was with the highest fold of change, which lies in the PI3K-AKT pathway (Fig. 2c) and the focal adhesion functional category (Fig. 2d and Supplementary Fig. 5c). Laminin α5 is one of the most widely distributed laminins in the epithelial basement membranes, which is involved in various biological activities such as cell adhesion, migration, growth, differentiation, and tumor metastasis[15]. However, it is unknown whether laminin α5 plays a role in IL-6-induced osimertinib resistance.

We next asked whether Laminin α5 was required for IL-6-induced osimertinib resistance. Laminin α5 was found over-expressed in osimertinib-resistant PC-9GROR cells, PC-9OR cells, and H1975OR cells, as compared to parental cells, respectively (Fig. 2e). Then, addition of IL-6 to sensitive PC-9 cells, PC-9GR cells, H1975 cells, HCC827 cells, or H3255 cells also increased Laminin α5 levels (Fig. 2e). Similarly, higher expression of Laminin α5 was found in IL-6 over-expressing IL-6-GFP-PC-9 cells and IL-6-GFP-PC-9GR cells (Fig. 2e). These data indicated that IL-6 can increase Laminin α5 expression.

Next, we studied the role of Laminin α5 in osimertinib resistance. Small interfering RNA (siRNA)-mediated knockdown of LAMA5 in PC-9GROR cells resulted in the re-sensitization of these resistant cells to osimertinib, together with inhibition of phosphorylated STAT3 (Fig. 2f, g). Moreover, in IL-6-GFP-PC-9 cells, IL-6-GFP-PC-9GR cells, and IL-6-GFP-HCC827 cells, siRNA knockdown of LAMA5 also enhanced cytotoxicity of osimertinib (Fig. 2f, g; Supplementary Fig. 6a, b). Taken together, these data suggest that Laminin α5 was required for IL-6-induced osimertinib resistance.

**Laminin α5 mediated IL-6-induced osimertinib resistance through regulation of FAK phosphorylation.** To investigate how Laminin α5 mediates IL-6-induced osimertinib resistance, we built a sub-network of PPI on DEPs of the afore-mentioned four IL-6 downstream pathways, based on proteomics data of PC-9GR cells and PC-9GROR cells. LAMA5 was shown to interact with LAMB1, LAMC1, PTK2, PRKCA, PAK1, and PAK2. Of note, PTK2, protein tyrosine kinase 2, also called focal adhesion kinase (FAK), was the protein with the highest degree of connectivity in this sub-network (Fig. 3a). FAK is a non-receptor tyrosine kinase with key roles in the regulation of cell adhesion, migration, proliferation, and survival[16]. Moreover, it has been reported previously that FAK phosphorylation was increased after osimertinib treatment, while inhibition of FAK phosphorylation showed only a modest effect on osimertinib efficacy[17]. We, therefore, hypothesized that FAK phosphorylation by Laminin α5 plays a role in IL-6-induced osimertinib resistance. Higher expression of phosphorylated FAK was found in osimertinib-resistant cells, by either immunofluorescence staining or Western blot assay (Fig. 3b, c). Also, increased phosphorylation of FAK was found in osimertinib-sensitive PC-9 cells, PC-9GR cells, H1975 cells, HCC827 cells, and H3255 cells treated with IL-6, or in IL-6-GFP PC-9 cells and IL-6-GFP PC-9GROR cells (Fig. 3c). We then assessed whether FAK phosphorylation was controlled by Laminin α5. In IL-6-GFP-PC-9 cells, IL-6-GFP-PC-9GR cells, and IL-6-GFP-HCC827 cells, siRNA knockdown of LAMA5 decreased FAK phosphorylation (Fig. 3d and Supplementary Fig. 6c). We next asked whether FAK phosphorylation was required for IL-6-induced osimertinib resistance. SiRNA knockdown of PTK2 in PC-9GROR cells, IL-6-GFP-PC-9 cells, IL-6-GFP-PC-9GR cells, or IL-6-GFP-HCC827 cells increased osimertinib sensitivity of these resistant cells (Fig. 3e, f; Supplementary Fig. 6d, e). Then, we assessed the expression levels of IL-6, Laminin α5, and phosphorylated FAK in tumor tissues of a total of six patients with osimertinib resistance. Strong staining of IL-6, Laminin α5, and phosphorylated FAK was found in patients with high plasma IL-6 levels upon osimertinib resistance ($N = 3$), while low expression of these proteins was found in those with low plasma IL-6 levels ($N = 3$, representative images shown in Fig. 3g). Taken together, these data suggest that Laminin α5/FAK pathway mediated IL-6-induced osimertinib resistance.

**Laminin α5/FAK signaling specifically mediated IL-6-induced osimertinib resistance.** We then asked whether Laminin α5/FAK signaling specifically mediated IL-6-induced osimertinib resistance. Single-cell clones were isolated and expanded in PC-9OR cells, PC-9GROR cells, and H1975OR cells, and a total of 13 monoclonal cell lines were generated, which were all highly resistant to osimertinib (Fig. 4a, b). Interestingly, increased IL-6

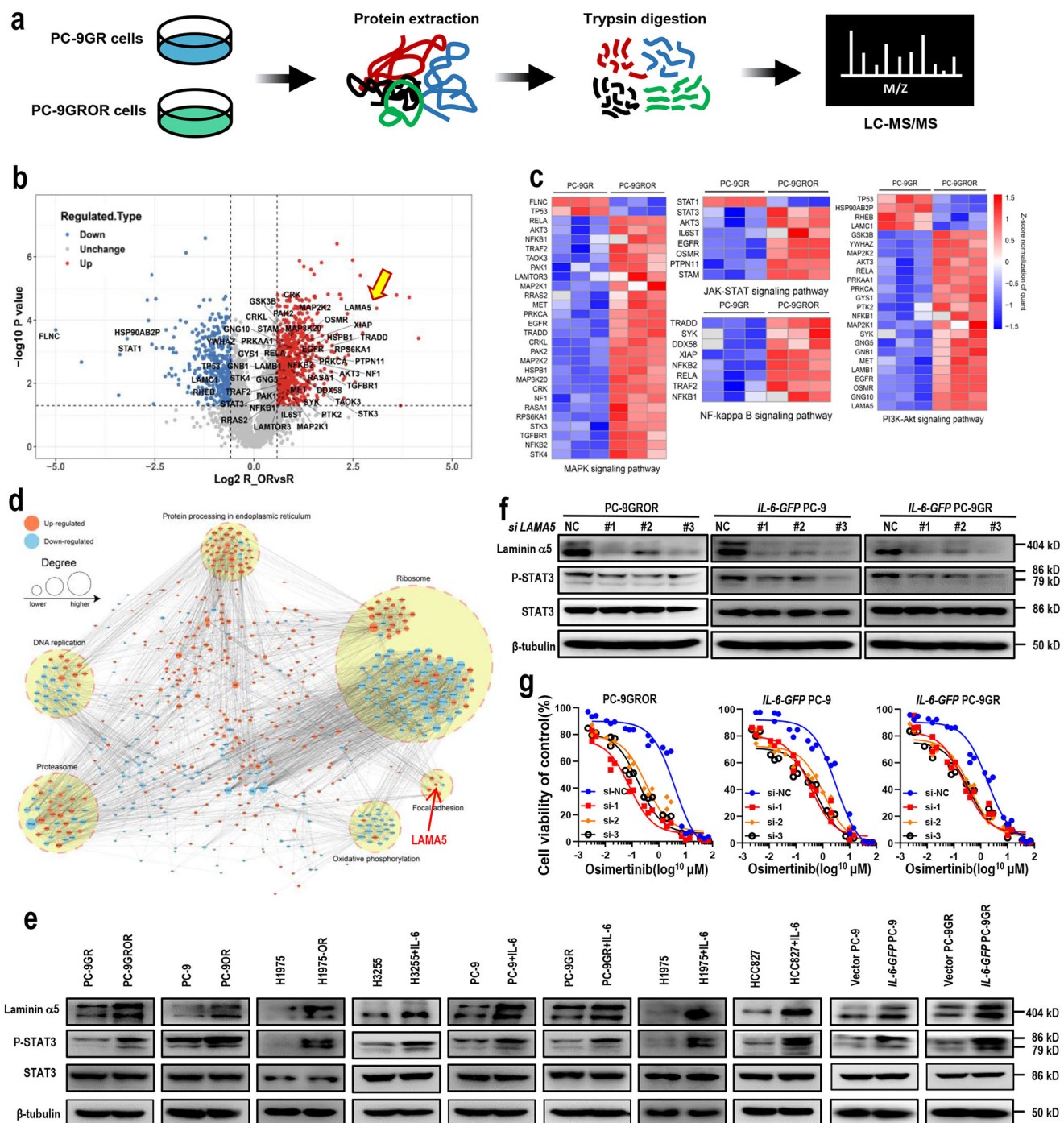

**Fig. 2 Laminin α5 was required for IL-6-induced osimertinib resistance. a** Experimental workflow of the proteomic experiments (biological triplicates per group). **b** Volcano plot of the differentially expressed proteins in PC-9GROR cells compared to PC-9GR cells, including significantly (FDR < 5%) upregulated (red dots), downregulated (blue dots), and unchanged (Gray dots) proteins. $X$ axis = $\log_2$ fold change; $Y$ axis = $-\log_{10}$ $p$-value. Cutoff of $p = 0.05$ and 1.5-fold change were marked by horizontal dashed line and vertical dashed line, respectively. Proteins in four IL-6 downstream pathways, MAPK signaling pathway, JAK-STAT pathway, NF-kB pathway, and PI3K-AKT pathway were marked, and Laminin α5 (LAMA5) was highlighted with an arrow. **c** Heat map of four indicated signaling pathways. Each row corresponds to a certain protein involved in the indicated pathway with its name in the left panel. The colors indicate regulation intensity (upregulated or downregulated using Z-score normalization of quant). **d** Network of interactions between proteins that are significantly (1.5 fold-change) regulated. Red nodes are upregulated proteins (>1.5-fold), and the blue nodes are downregulated proteins (>1.5-fold). Node size is proportional to the degree of protein-protein interaction. Greater the node is, more proteins interact with it. Each dashed circle represents a functional category. LAMA5 in the focal adhesion category was highlighted with an arrow. **e** Western blot showing the expression levels of certain proteins in cell lines as indicated. IL-6 (20 ng/ml) was added to the culture medium for 48 h. β-tubulin served as loading control. **f** The levels of Laminin α5 and STAT3 in PC-9GROR cells, *IL-6-GFP* PC-9 cells, and *IL-6-GFP* PC-9GR cells after transfection with *LAMA5* siRNAs, respectively. β-tubulin served as loading control. **g** Cell viability CCK-8 assay for indicated cells transfected with control or *LAMA5* siRNAs, respectively, and treated with increasing concentrations of osimertinib for 48 h. Data are shown as mean ± SEM ($n = 3$ biologically independent experiments).

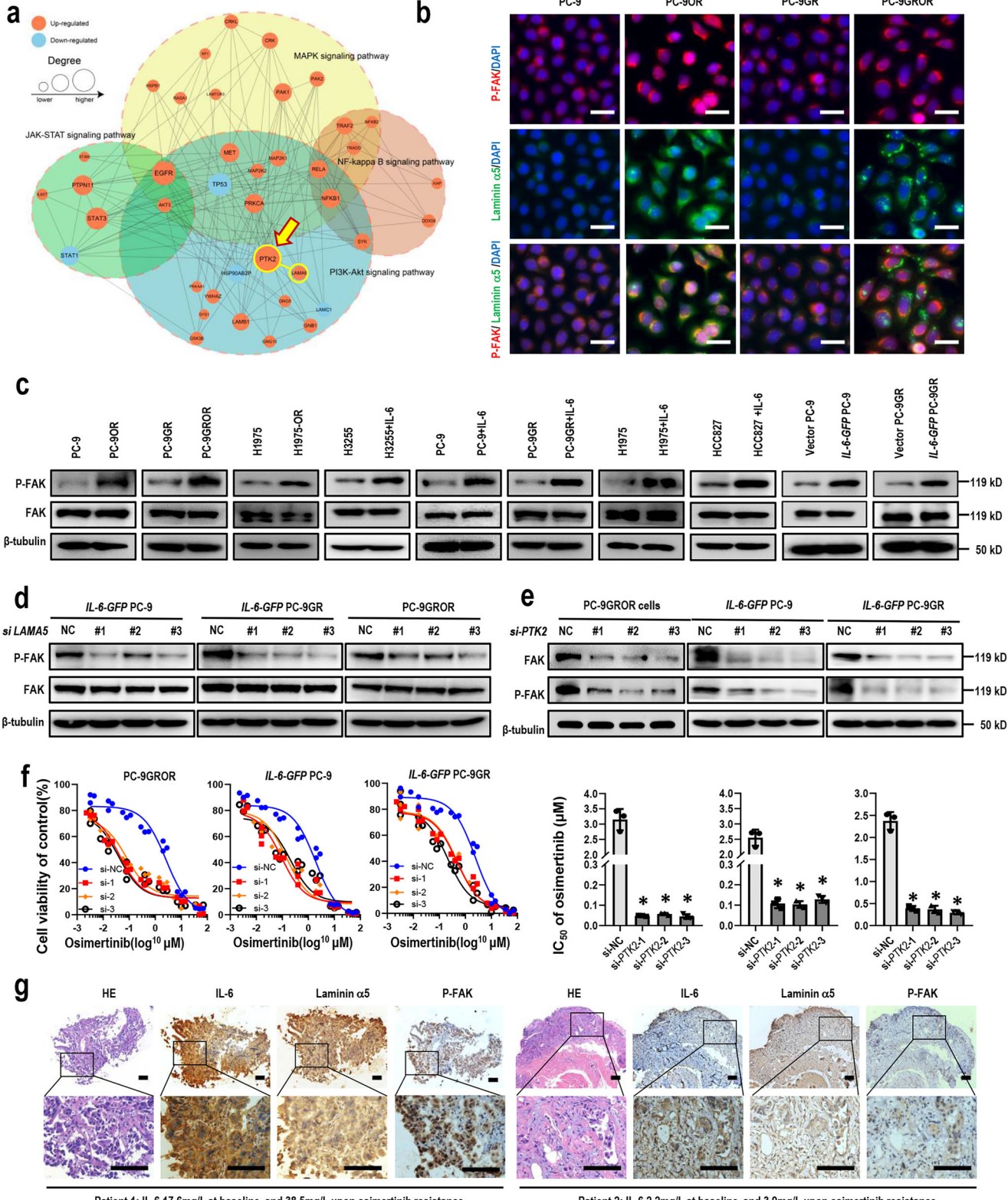

Patient 1: IL-6 17.6mg/L at baseline, and 38.5mg/L upon osimertinib resistance

Patient 2: IL-6 2.2mg/L at baseline, and 3.9mg/L upon osimertinib resistance

secretion was found in 12 of those cell lines, while the H1975OR7 cells expressed lower levels of IL-6 as compared to parental H1975 cells (Fig. 4c). Therefore, through single-cell clone selection, we established osimertinib-resistant cell lines with IL-6-dependent and IL-6-independent resistance mechanisms. Next, we studied the role of Laminin α5/FAK signaling in osimertinib resistance in different H1975ORs. As expected, Laminin α5 and

phosphorylated FAK were both overexpressed in PC-9OR cells, PC-9GROR cells, and in H1975 OR1, 2, 4, 5, 6 cells, while its level remained unchanged in H1975OR7 cells (Fig. 4d, e).

Next, we asked whether Laminin α5 or FAK was required for osimertinib resistance in H1975ORs with different IL-6 levels. Knockdown of *LAMA5* reduced levels of phosphorylated STAT3 and enhanced osimertinib cytotoxicity in H1975OR2 cells, but

**Fig. 3 Laminin α5 mediated IL-6-induced osimertinib resistance through phosphorylation of FAK. a** Protein–protein network of the four indicated signal pathways. Red nodes are upregulated proteins (>1.5-fold), and the blue nodes are downregulated proteins (>1.5-fold). Node size is proportional to the degree of protein–protein interaction. The dashed circles frame the corresponding pathways. LAMA5 and PTK2 were highlighted with circles. **b** Representative images of Laminin α5 and phosphorylated FAK in indicated cell lines by fluorescent detection. Cells were counterstained with DAPI. Scale bars: 30 μm. **c** Western blot showing the expression levels of FAK and phosphorylated FAK in cell lines as indicated. IL-6 (20 ng/ml) was added to the culture medium for 48 h. Experiments were performed in triplicates, and β-tubulin served as loading control. **d** Western blot showing the levels of total and phosphorylated PTK2 in PC-9GROR cells after transfection with *LAMA5* siRNAs, respectively. β-tubulin served as loading control. **e** The levels of total and phosphorylated FAK in PC-9GROR cells after transfection with *PTK2* siRNAs, respectively. β-tubulin served as loading control; **f** cell viability CCK-8 assay for PC-9GROR cells transfected with control or *PTK2* siRNAs, respectively, and treated with increasing concentrations of osimertinib for 48 h. Data are shown as mean ± SEM ($n = 3$ biologically independent experiments). Histogram shows $IC_{50}$ values in the indicated groups (*$p < 0.01$ by Student's t-test). **g** Immunohistochemistry analysis of IL-6, Laminin α5, and phosphorylated FAK on tumor sections from two representative patients upon osimertinib resistance. HE, Hematoxylin and Eosin staining. Scale bars: 100 μm.

showed little effect on phosphorylated STAT3 expression or osimertinib sensitivity in H1975OR7 cells (Fig. 4f, g). Similarly, after siRNA-mediated knockdown of *PTK2*, STAT3 phosphorylation was suppressed in H1975OR2 cells, along with enhanced osimertinib sensitivity. On the other hand, knockdown of *PTK2* showed little effect in STAT3 activation or osimertinib sensitivity in H1975OR7 cells with low levels of IL-6 (Fig. 4h, i). Taken together, these results suggest that Laminin α5/FAK signaling specifically mediated IL-6-induced osimertinib resistance.

**Compound screening revealed that ibrutinib could overcome IL-6-induced osimertinib resistance through inhibition of Laminin α5/FAK signaling.** To discover small molecules that target IL-6/Laminin α5/FAK signaling in lung cancer, we first performed compound screens in two cell lines (*IL-6-GFP* PC-9 cells and *IL-6-GFP* PC-9GR cells) using a small molecule library comprising 510 compounds (Fig. 5a and Supplementary Data 3). Among the top 20 small molecules that decreased IL-6 levels in each cell line, a total of five common compounds were selected, including brigatinib, ibrutinib, daunorubicin, vandetanib, and ceritinib (Fig. 5b, c and Supplementary Data 3). To validate the results of the screen, we performed an ELISA assay and confirmed that all those five compounds decreased IL-6 levels in both cell lines (Fig. 5d). Next, we performed secondary screens of those five compounds using Western Blot to detect the activation of IL-6/Laminin α5/FAK pathway. We found that ibrutinib, a small molecule inhibitor targeting Bruton's tyrosine kinase (BTK)[18], a key non-receptor tyrosine kinase in the B-cell receptor signaling pathway, consistently and effectively suppressed the levels of phosphorylated STAT3, phosphorylated FAK, and Laminin α5 across both cell lines (Fig. 5e). Taken together, these cell-based screening experiments identified ibrutinib as a potent inhibitor of IL-6/ Laminin α5/FAK signaling.

We then asked whether ibrutinib could overcome IL-6-induced osimertinib resistance in vitro and in vivo. Ibrutinib effectively enhanced osimertinib cytotoxicity in *IL-6-GFP* PC-9 cells, *IL-6-GFP* PC-9GR cells, and *IL-6-GFP* HCC827 cells, as well as in osimertinib-resistant PC-9OR3 cells and PC-9GROR3 cells, while ibrutinib alone resulted in only slight decrease of cell viability (Fig. 6a and Supplementary Fig. 6f, g). Also, ibrutinib together with osimertinib decreased the levels of phosphorylated STAT3, phosphorylated FAK, and Laminin α5, in the afore-mentioned cell lines (Fig. 6b and Supplementary Fig. 6h). More importantly, ibrutinib was able to reverse osimertinib resistance and suppress Laminin α5/FAK signaling and STAT3 activation in H1975-OR6 cells (with high IL-6 expression), while showing little effect in H1975-OR7 cells (with low IL-6 expression), thus indicating that ibrutinib specifically overcome IL-6-induced osimertinib resistance (Fig. 6a, b). We then assessed the role of BTK in ibrutinib's activity on osimertinib sensitivity and IL-6 levels. SiRNA knockdown of *BTK* in PC-9GROR cells or *IL-6-GFP*-PC-9GR

cells increased osimertinib sensitivity, and decreased IL-6 levels. Interestingly, after siRNA-mediated knockdown of *BTK*, ibrutinib failed to further increase osimertinib sensitivity or decrease IL-6 levels (Supplementary Fig. 7). Next, the efficacy of ibrutinib in combination with osimertinib was further assessed in PC-9GROR xenografts. Treatment with osimertinib or ibrutinib alone had little effect on tumor size, while the combination resulted in tumor shrinkages (Fig. 6c, d), along with decreased IL-6 levels (Fig. 6e). Taken together, these findings suggest that ibrutinib enhanced the therapeutic efficacy of osimertinib in *vitro* and in *vivo*, possibly through inhibition of Laminin α5/FAK signaling pathway.

**Discussion**
The long-term effectiveness of osimertinib is limited by acquired resistance. IL-6 has been indicated as a resistance mechanism to 1st and 2nd generation EGFR-TKIs, yet its potential role in osimertinib resistance was unknown. In the current study, we found that IL-6 mediated osimertinib resistance through Laminin α5/ FAK signaling activation, which can be overcome by ibrutinib, identified by compound screening (Fig. 6f). Our findings suggest that ibrutinib in combination with osimertinib may be developed as a promising strategy to overcome IL-6 induced osimertinib resistance.

IL-6 has been implicated as a driver of 1st or 2nd generation EGFR TKI resistance and associated with a worse outcome in TKI-treated NSCLC patients[11,12]. IL-6 was suggested to be associated with TKI resistance of unknown mechanisms, and its expression was minimal in cells where EGFR TKI resistance was associated with MET amplification[11]. In the current study, analysis of clinical data sets revealed that pretreatment high circulating levels of IL-6 were associated with a worse PFS in patients treated with osimertinib. IL-6 levels were further increased in patients with osimertinib resistance, among whom low IL-6 levels were associated with known osimertinib resistance mechanisms such as C797S mutation or MET amplification, while high IL-6 levels were associated with a higher proportion of yet unknown resistance mechanisms. Given the feasibility and wide application of IL-6 testing, we propose that monitoring of serum IL-6 levels can be used for the detection of osimertinib resistance. However, one should bear in mind that blood IL-6 could be derived from multiple types of cells other than cancer cells, such as B cells, macrophages, and fibroblasts[19–21]. Thus, a comprehensive analysis is needed to correlate IL-6 increase with osimertinib resistance.

Previously, IL-6 was shown to mediate resistance to 1st generation EGFR-TKI resistance through activation of STAT3, which can be blocked by IL-6 neutralizing antibodies[13]. However, in the current study, we found that either a neutralizing IL-6R antibody tocilizumab or a STAT3 inhibitor STATTIC failed to overcome IL-6-induced osimertinib resistance, indicating that IL-6 induced

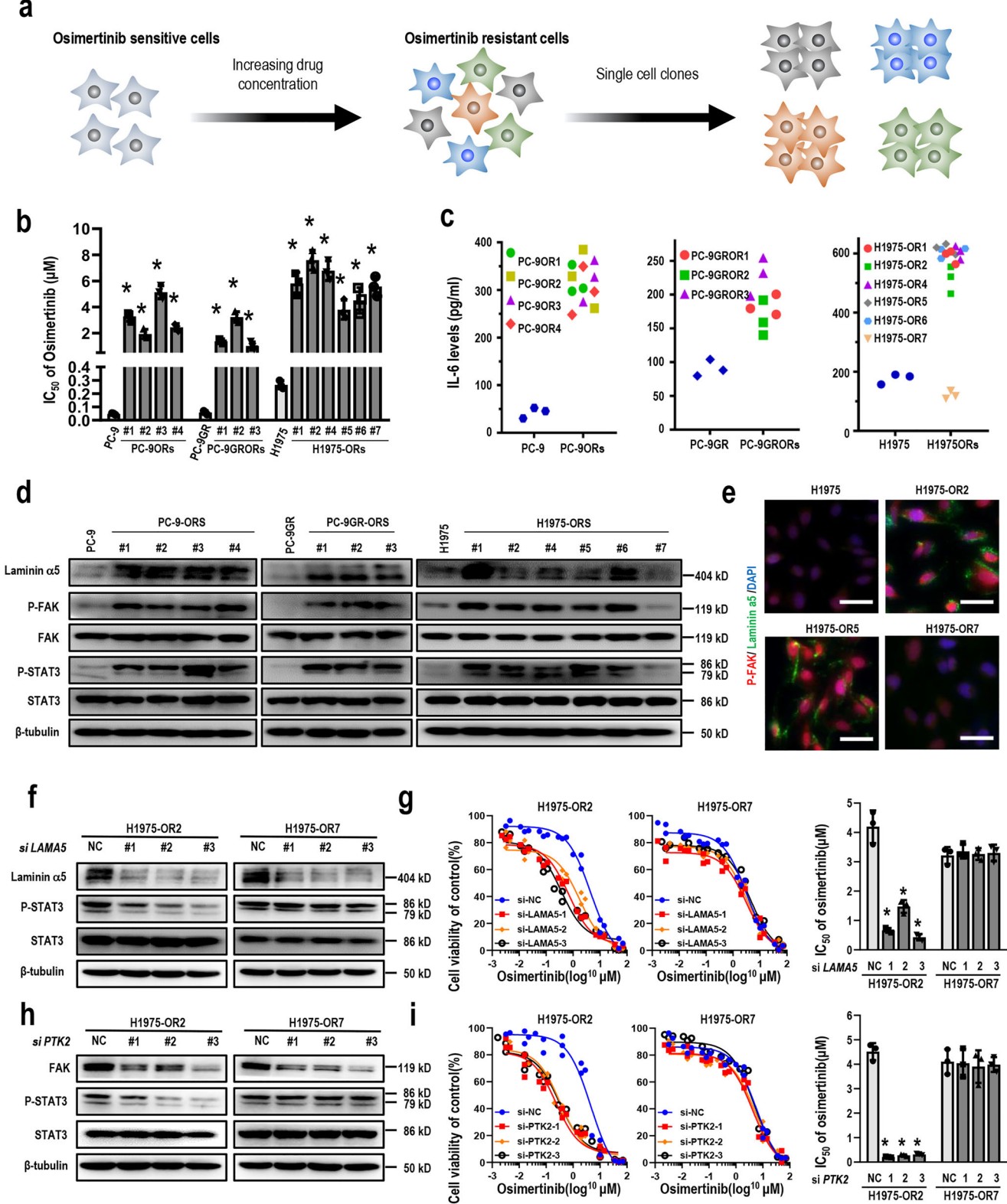

osimertinib resistance through more complex mechanisms. As described earlier, activation of multiple receptor tyrosine kinases can result in sustained feedback activation of STAT3 in cancer drug resistance[22]. Thereafter, we performed proteomics and found that MAPK, JAK-STAT, NF-kB, and PI3K-AKT pathways were all highly activated in osimertinib-resistant cells, and laminin α5 was with the highest fold of upregulation. Laminin α5 is one of the most widely distributed laminins in the developing and mature organism, and is also a major component of extracellular membrane (ECM)[23]. Previously, it was reported that the ECM alone can induce tumor cell resistance to treatment[24], and IL-6 can regulate ECM remodeling[25,26]. In the current study, laminin α5 was found overexpressed in tumor tissues of patients with osimertinib resistance and high plasma IL-6 levels, as well as in osimertinib-resistant cell lines with high IL-6 levels. Further experiments proved that laminin α5 specifically mediated

**Fig. 4 Laminin α5/FAK pathway specifically mediated IL-6-induced osimertinib resistance. a** Schematic overview of generation of osimertinib-resistant single clones. Increasing concentrations of osimertinib was applied to osimertinib-sensitive cells to generate resistant cells. Then, limiting dilution method was applied and osimertinib-resistant single-cell clones were further selected; **b** histogram showing IC$_{50}$ values of the selected osimertinib-resistant single-cell clones (*$p < 0.01$ compared with parental cells, respectively). $n = 3$ biologically independent experiments; **c** IL-6 levels in osimertinib-resistant single-cell clones detected by ELISA. **d** Western blot showing the expression levels of Laminin α5, total and phosphorylated FAK in indicated cell lines. β-tubulin served as loading control; **e** representative images of Laminin α5 and phosphorylated FAK in indicated cell lines by fluorescent detection. Cells were counterstained with DAPI. Scale bars: 50 μm. **f** Western blot showing the levels of Laminin α5, total and phosphorylated STAT3 in H1975OR2 cells and H1975OR7 cells transfected with control or *LAMA5* siRNAs. β-tubulin served as loading control. **g** Cell viability CCK-8 assay for H1975OR2 cells and H1975OR7 cells transfected with control or *LAMA5* siRNAs, respectively, and treated with osimertinib for 48 h ($n = 3$ biologically independent experiments). Histogram shows IC$_{50}$ values in the indicated groups (*$p < 0.01$ by Student's t-test). **h** The levels of FAK, total and phosphorylated STAT3 in H1975OR2 cells and H1975OR7 cells after transfection with *PTK2* siRNAs, respectively. β-tubulin served as loading control. **i** Cell viability CCK-8 assay for indicated cells transfected with control or *PTK2* siRNAs, respectively, and treated with osimertinib for 48 h ($n = 3$ biologically independent experiments). Histogram shows IC$_{50}$ values in the indicated groups (*$p < 0.01$ by Student's t-test).

IL-6-induced osimertinib resistance. Therefore, our results suggested that IL-6 remodeled ECM by upregulating laminin α5, which resulted in osimertinib resistance.

In the current study, FAK was shown to be a potential downstream target of laminin α5. FAK is a cytoplasmic tyrosine kinase that is overexpressed and activated in several advanced-stage solid cancers, which is at the intersection of various signaling pathways that promote cancer growth, invasion, gene expression, and cancer stem cell self-renewal[27,28]. Previously, it was reported that upon cell adhesion to the ECM, activated integrins led to an increase in the phosphorylation of FAK[29]. In A549 cells, cell adhesion to laminin-10/11 mediated by α3β1 integrins also led to phosphorylation of FAK and activation of Rho family of GTPases[30]. We have also proved that FAK activation was modulated by laminin α5, indicating that over-expressed laminin α5 induced osimertinib resistance through FAK activation. Several studies have reported that activation of FAK signaling is associated with EGFR-TKI resistance. FAK/AKT and ERK activation by osteopontin contributes to acquired resistance of gefitinib[31]. In a pemetrexed-resistant NSCLC cell line PC-9/PEM, which also acquired EGFR-TKI resistance, FAK was hyperphosphorylated and a combination of FAK inhibitor and osimertinib recovered EGFR-TKI sensitivity[32]. However, another study found that although osimertinib treatment resulted in autophosphorylation of FAK, the combination of FAK inhibitor PF573228 and osimertinib only had modest effects on cell viability[17]. In the current study, FAK activation was found only in osimertinib-resistant cell lines with high IL-6 levels, but not in those with low IL-6 levels. Moreover, knockdown of *PTK2* restored osimertinib sensitivity in IL-6-overexpressing cells and osimertinib-resistant cell lines with high IL-6 levels, but not in those with low IL-6 levels. As reported previously, activation of FAK was required for IL-6 production in mesenchymal stromal stem cell maintenance[33]. Taken together, our results suggest that FAK activation was specifically related to IL-6-induced osimertinib resistance.

Ibrutinib (PCI-32765) is an orally available inhibitor of the BTK and has been approved for the treatment of several different types of hematological malignancies, such as chronic lymphocytic leukemia, mantle cell lymphoma, or Waldenström's macroglobulinemia[34]. It has been suggested that ibrutinib exhibits anti-tumorigenic functions in preclinical models of a variety of solid tumors, such as pancreatic, breast, lung, gastric, and ovary cancer[35,36]. Ibrutinib has demonstrated pre-clinical efficacy in EGFR mutant NSCLC, with high activity against EGFR mutated (L858R) cells and moderate activity to T790M mutated cells[37]. Currently, Ibrutinib is undergoing evaluation in a phase I/II trial in previously-treated EGFR-mutant NSCLC (clinicaltrials.gov ID: NCT02321540). However, one previous study found that Ibrutinib inhibits mutant EGFR kinase through formation of a covalent bond with Cys797, yet this irreversible binding is much less efficient than other irreversible EGFR inhibitors such as WZ4002 or AZD9291 (osimertinib), due to an unusual DFG-in/c-Helix-out binding conformation[38]. Therefore, ibrutinib as a single agent to treat EGFR-mutant NSCLC is facing an inevitable dilemma. In the current study, we also noticed that ibrutinib monotherapy showed little effect on tumor growth in PC-9GROR xenografts. On the other hand, when combined with osimertinib, ibrutinib selectively overcame drug resistance in osimertinib-resistant cell lines with high IL-6 levels, not in those with low IL-6 levels, through inhibition of IL-6 and Laminin α5/FAK signaling. Our findings, together with those previous studies, suggest that ibrutinib may not be a good choice in unselected EGFR-mutant NSCLC, but hold the promise in a subgroup of patients with osimertinib resistance induced by IL-6.

Taken together, we conclude that Laminin α5/FAK signaling activation mediates IL-6-induced osimertinib resistance, and ibrutinib is a promising therapeutic strategy to overcome IL-6-mediated osimertinib resistance.

## Methods

**Experimental design.** We retrospectively assessed patients with histologically-confirmed advanced EGFR-mutant NSCLC who had been treated with EGFR-TKIs at Daping Hospital, Army Medical University, from March 1st 2014 to Dec 31th 2020. Patients with baseline blood IL-6 levels were included, and those with unknown mutational status, or confirmed bacterial infection, or those without response evaluation were excluded. The following information was extracted from medical charts: age, gender, disease stage, tumor histology, type of EGFR mutations, radiological examinations, and blood tests for IL-6 and WBC. EGFR mutation status was determined either by tissue-based amplification refractory mutation system-polymerase chain reaction (ARMS-PCR) or next-generation sequencing (NGS). Serum IL-6 level was retrieved from medical records, which had been measured by Enzyme-linked immunosorbent assay (ELISA) at Department of Clinical Laboratory, Daping Hospital in clinical tests. The study was approved by the Ethics Committee of Daping Hospital, Army Medical University (NO. 2020143) and was conducted according to the Declaration of Helsinki. Trial Registration: ChiCTR2100046643. Informed consents were obtained from all patients or their legal guardians.

**Cell lines and reagents.** Human lung cancer cell lines PC-9 cells, H1975 cells, HCC827 cells, and H3255 cells were from the American Type Culture Collection (ATCC). PC-9GR cells were gifted by Prof. J. Xu and Dr. M. Liu from Guangzhou Medical University (China). Osimertinib resistant cell lines were established as previously reported[39]. Briefly, cells were first treated with osimertinib at the concentration of IC$_{50}$ for 2 weeks, and then treated with increasing concentrations of osimertinib, until resistant variants emerged. For generation of osimertinib-resistant single cell clones, the resistant cells were seeded single cell per well using limiting dilution, and cultured continuously in the presence of osimertinib. Cell lines were cultured in RPMI-1640 (Hyclone) with Earle's salts, supplemented with 10% FBS (Gibco), 2 mmol/L L-glutamine (Gibco), 100 U/ml penicillin (HyClone), and 100 μg/mL streptomycin (Hyclone) at 37 °C, with 5% CO$_2$ and 90% humidity.

Osimertinib (Tagrisso) was provided by Astra Zeneca. The recombinant IL-6 was purchased from PeproTech (New Jersey, USA). Ibrutinib (S2680), and tocilizumab (A2012) were from Selleck (Texas, USA). Antibodies against phospho-(Tyr705)-STAT3(#9145), STAT3(#12640) were from Cell Signaling Technology (Massachusetts, USA). Antibodies against phospho-(Y397)-FAK (ab81298), FAK

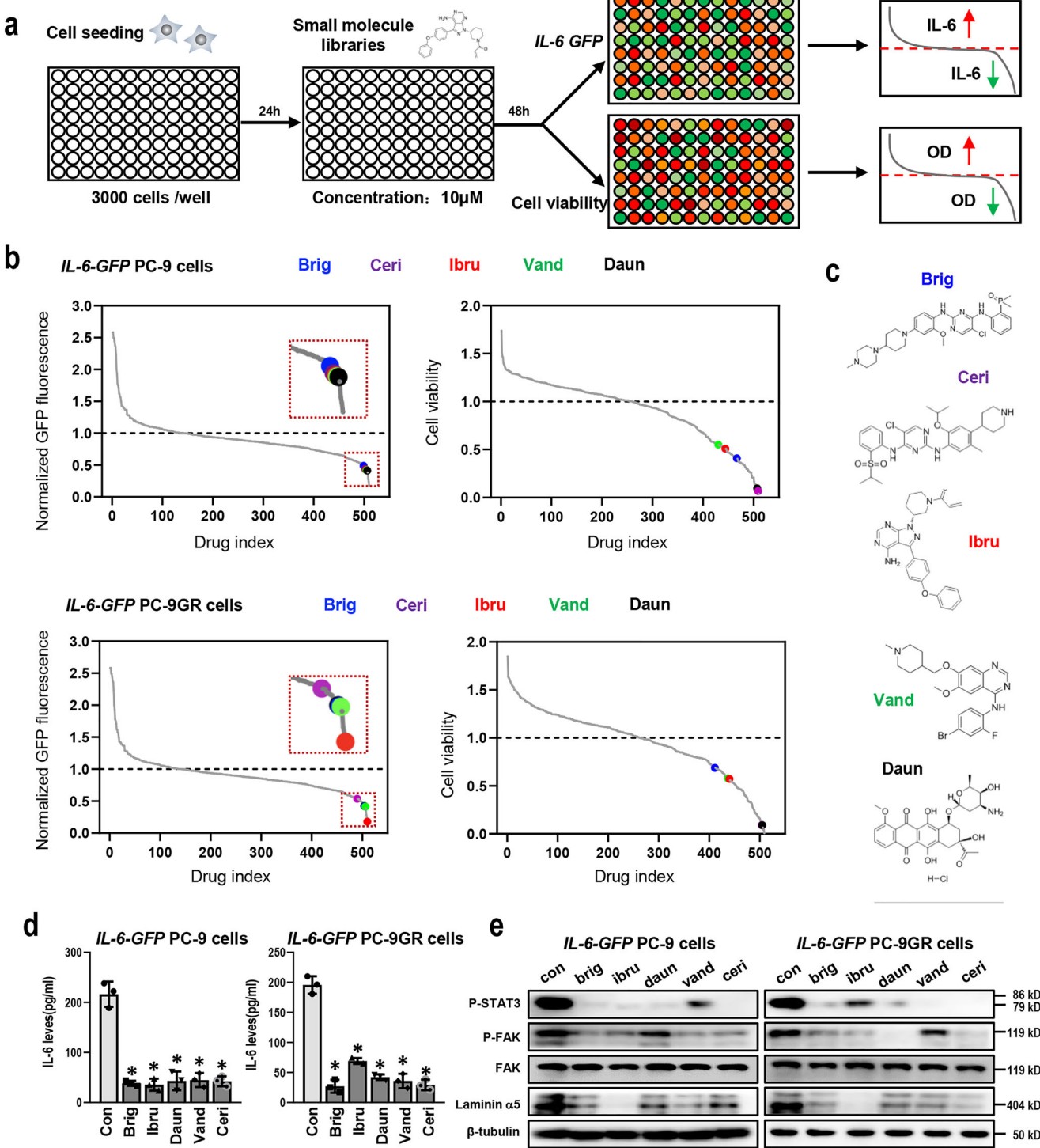

**Fig. 5 Compound screening identified ibrutinib as a potent inhibitor of IL-6 and Laminin α5/FAK pathway. a** Schematic overview of the screening procedure. Lung cancer cell lines stably expressing GFP-labeled IL-6 were seeded onto 96-well plates in two separate sets. After 24 h, compound libraries (containing 510 drugs) were added and cells were treated for another 48 h, followed by measurement of cell viability in one set and GFP levels in the other set of plates ($n = 2$ biologically independent experiments for each screening); **b** waterfall plots showing the effect of the compound library on GFP levels and cell viability in *IL-6-GFP* PC-9 cells and *IL-6-GFP* PC-9GR cells. The mean value of two independent experiments was presented. The selected five compounds are shown as colored dots as indicated. Brig, brigatinib; Ceri, ceritinib; Ibru, ibrutinib; Vand, vandetanib; Daun, daunorubicin; **c** chemical structure of selected five compounds: Brigatinib (PubChem CID 68165256), Ceritinib (PubChem CID 57379345), ibrutinib (PubChem CID 24821094), Vandetanib (PubChem CID 308136), Daunorubicin (PubChem CID 62770); **d** ELISA assay of IL-6 levels in cells treated with different compounds as indicated ($n = 3$ biologically independent experiments). Data are shown as mean ± SEM. *, $p < 0.01$ as compared to control. **e** Western blot showing the expression levels of indicated proteins in *IL-6-GFP* PC-9 cells and *IL-6-GFP* PC-9GR cells in the presence of five selected compounds. β-tubulin served as loading control.

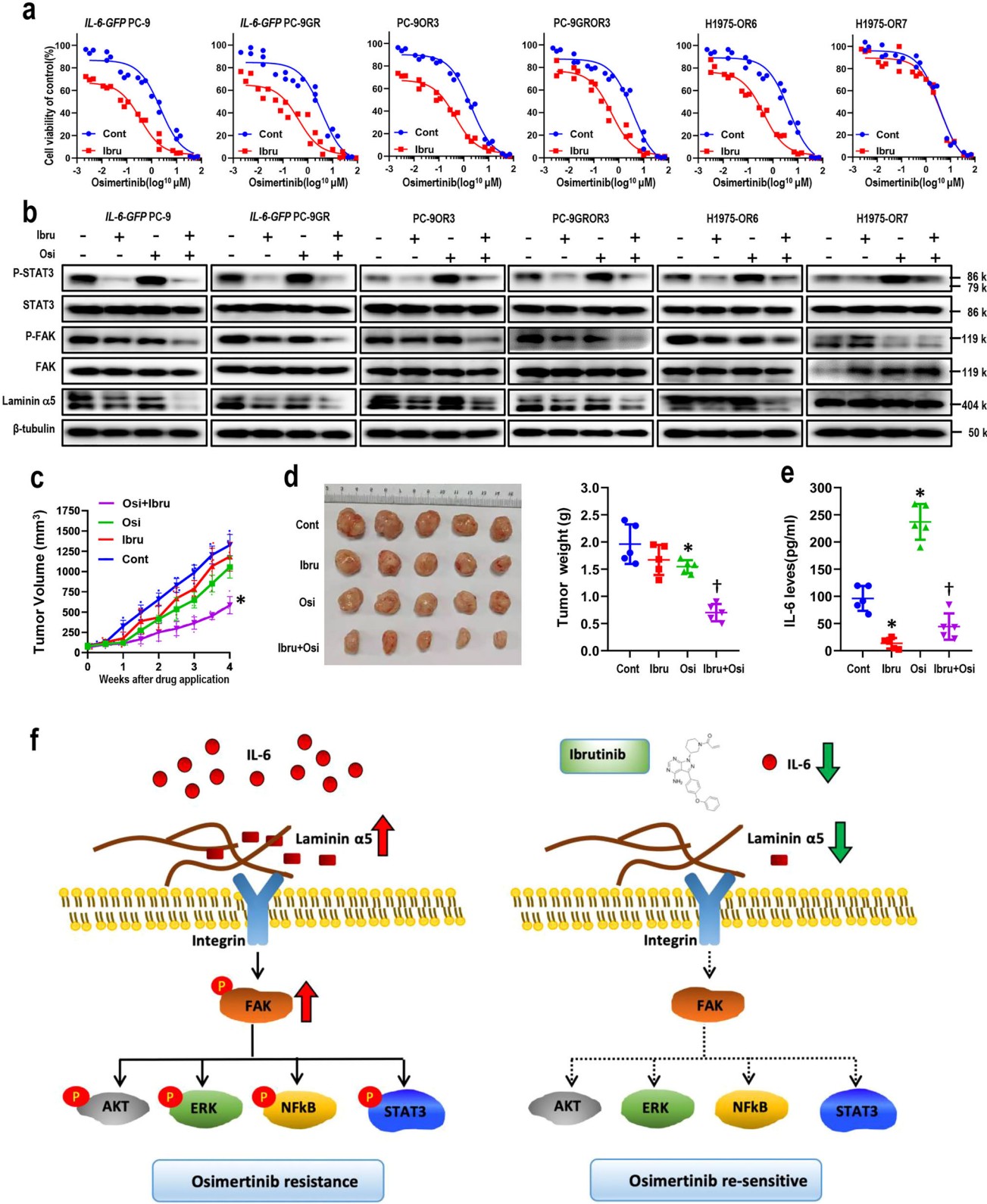

(ab40794), Laminin α5 (LAMA5, #ab77175, #ab184330) were from Abcam (Cambridge, UK). β-tubulin (#A5032) was from Bimake (Texas, USA). The concentrations of antibodies were listed in Supplementary Table 2.

**Elisa assay**. Quantitative detection of IL-6 was determined by IL-6 ELISA kit (solarbio, Beijing, China). The assay was performed in triplicates, following the manufacturer's instructions. The plate was read by Thermo Labsystems Multiskan Ascent Photometric plate reader for 96 well plates.

**Lentivirus production and transduction**. To generate cell lines overexpressing IL-6, the human IL-6 cDNA sequence (Genebank accession number: NM_000600) was searched for suitable target sequences. Lenti-*IL-6-GFP* and Lenti-NC virus were designed and generated by Gene-Chem Co., Ltd (Shanghai, China). DNA oligos containing the target sequence (forward primer: AGGTCGACTCTAGAG GATCCCGCCACCATGAACTCCTTCTCCACAAG, reverse primer: TCCT TGTAGTCCATACCCATTTGCCGAAGAGCC) were inserted into the GV492 vector by double digestion with BamHI and AgeI. PC-9 cells, PC-9GR cells, and HCC827 cells were transfected with ViraPower packaging mix using Lipofectamine

**Fig. 6 Ibrutinib specifically overcame IL-6-induced osimertinib resistance in vitro and in vivo. a** Cell viability CCK-8 assay for multiple cell lines treated with ibrutinib (0.5 μM) and increasing concentrations of osimertinib for 48 h ($n = 3$ biologically independent experiments). Data are expressed as mean ± SEM. **b** Western blot showing the expression levels of indicated proteins in multiple cell lines in the presence of ibrutinib, osimertinib, or both. β-tubulin served as loading control; **c** tumor growth of PC-9GROR xenografts treated with osimertinib, ibrutinib, or the combination. Tumor sizes were presented as mean ± SEM ($n = 5$ biologically independent animals). *, $p < 0.01$ as compared to control, ibrutinib alone or osimertinib alone. **d** Macroscopic appearance and tumor weights of the xenografts in different groups as indicated. *, $p < 0.05$ as compared to control; †, $p < 0.01$ as compared to control, ibrutinib alone or osimertinib alone. **e** Plasma IL-6 levels of mice from the four groups determined by ELISA. The data are expressed as mean and interquartile range. *, $p < 0.01$ as compared to control; †, $p < 0.01$ as compared to osimertinib alone. **f** Schematic overview of the findings of the current study.

2000 reagent according to manufacturer's instructions. Increased IL-6 levels in these generated cells were confirmed by ELISA.

**siRNA transfection.** Small interfering RNAs (siRNAs, RiboBio, Guangzhou, China) were used to silence endogenous *LAMA5* expression (sequence: GCA TCAGCTTCGACAGTCA; GCCTCGTGCTGTTGTATGA; TCACCGGACTG ATCTTCCA) or *PTK2* expression (sequence: GAACGGGCTTTGCCATCAA; CCCTAACCATTGCGGAGAA; GGACATTATTGGCCACTGT) or *BTK* expression (sequence: GAAGGAGGTTTCATTGTCA; GTAAGAAGGGTTCAATAGA; GCTCAAATATCCAGTGTCT) in PC-9GROR, *IL-6-GFP* PC-9 cells and *IL-6-GFP* PC-9GR cells. For the evaluation of efficacy, cells were transfected with either 80 pmol siRNA or negative control siRNA (siNC) using Lipofectamine RNAiMAX (Thermo Fisher Scientific, MA, USA), following the manufacturer's instructions. At 72 h post-transfection, knockdown efficiency was determined by examining endogenous protein expression by Western blot.

**RNA sequencing.** Transcriptome analysis was performed in paired osimertinib sensitive/resistant cells with biological triplicates as follows. Briefly, cell pellet was prepared and resuspended in TRIzol (Invitrogen). Following the addition of 200 μL chloroform, samples were vortexed and centrifuged at $13,000 \times g$ for 10 min at 4 °C. Then the total RNA was extracted using PureLink RNA Micro Kit (Thermo Fisher, 12183016) according to the manufacturer's protocol. The quality and integrity of RNA solution were verified using a NanoDrop 2000 spectrophotometer (Thermo Fisher Scientific) and Agilent Bioanalyzer 2100 (Agilent Technologies). Then the RNAseq libraries were constructed using the TruSeq Stranded mRNA LTSample Prep Kit (Illumina, San Diego, CA, USA) and sequenced on the Illumina sequencing platform (HiSeq™ 2500) and the final library size is a band 200–400 bp. Raw reads were trimmed for quality and aligned with the hg19 human genome using TopHat2[40]. Differentially expressed genes (DEGs) were identified using R-package CORNAS[41]. A *P* value < 0.05 and fold change > 2 or fold change < 0.5 were set as the threshold for significant differential expression. GO enrichment and KEGG pathway enrichment analysis of the DEGs were performed using R package clusterProfiler[42]. The PPIs were downloaded from String (https://string-db.org/)[43]. Network visualization and analysis were carried out by Cytoscape[44].

**Proteomics analysis.** Proteins from biological triplicates of each cell sample were extracted. Sample was sonicated three times on ice using a high intensity ultrasonic processor (Scientz) in lysis buffer (8 M urea, 1% Protease Inhibitor Cocktail). The remaining debris was removed by centrifugation at $12,000 \times g$ at 4 °C for 10 min. Finally, the supernatant was collected and the protein concentration was determined with BCA kit according to the manufacturer's instructions. Subsequently, 1000 μg of protein from each sample was digested with trypsin. The tryptic peptides were fractionated into fractions by high pH reverse-phase HPLC using Thermo Betasil C18 column (5 μm particles, 10 mm ID, 250 mm length), dissolved in solvent A (0.1% formic acid, 2% acetonitrile/ in water), and then loaded onto a home-made reversed-phase analytical column (25 cm length, 100 μm i.d.). The gradient was comprised of an increase from 6 to 24% solvent B (0.1% formic acid in 90% acetonitrile) over 70 min, 24–35% in 14 min and climbing to 80% in 3 min then holding at 80% for the last 3 min, all at a constant flow rate of 450 nL/min on a nanoElute UHPLC system (Bruker Daltonics). The peptides were subjected to Capillary source followed by the timsTOF Pro (Bruker Daltonics) mass spectrometry coupled online to the UPLC. Precursors and fragments were analyzed at the TOF detector. The electrospray voltage applied was 1.6 kV. The timsTOF Pro was operated in parallel accumulation serial fragmentation (PASEF) mode. Precursors with charge states 0–5 were selected for fragmentation, and 10 PASEF-MS/MS scans were acquired per cycle. The dynamic exclusion was set to 30 s. For bioinformatic analyses, upregulated or downregulated proteins (fold change ≧1.5 or ≤0.667, respectively, t-test $p < 0.05$) were defined as differentially expressed proteins (DEPs). The proteins were functionally annotated according to biological processes, molecular functions and cellular components derived from the UniProt-GOA database. All DEPs database accession or sequence were searched against the STRING database version 11.0 for PPIs. Only interactions between the proteins belonging to the searched data set were selected, thereby excluding external candidates. The interaction network from STRING was visualized in Cytoscape.

A graph theoretical clustering algorithm, molecular complex detection (MCODE) was utilized to analyze densely connected regions

**Cell viability assay.** Cell viability was determined by a 4-[3-(2-methoxy-4-nitro-phenyl)−2-(4-nitrophenyl)−2H-5-tetrazolio]-l,3-benzene disulfonate sodium salt (WST-8) assay using Cell Counting Kit-8 (CCK-8; MedChemExpress, New Jersey, USA). The $IC_{50}$ was calculated using Prism v. 8.00 (GraphPad Software, CA, USA). Cells were seeded in a 96-well plate at a density of 3000 per well and cultured with the indicated doses of drug-containing medium. Absorbances were measured on a Sunrise R microplate reader (Thermo Fisher Scientific, Germany) at 450 nm. The absorbance for the blank well was subtracted from each absorbance value. The absorbance of each well was expressed as a percentage of growth relative to the untreated cells to determine the relative cell viability percentage.

**Ki67 incorporation assay.** Cell proliferation was assessed by the Ki67 incorporation assay with a Ki67 labeling and detection kit (M00254-8, Boster, Wuhan, China). Briefly, cells seeded in six-well plates were treated with drug-containing medium as indicated for 48 h. Then cells were fixed and incubated overnight with Ki67 (1:200 dilution). Cells were counterstained with 4′, 6-diamidino-2-phenylindole (DAPI) for 15 min and observed under a fluorescence microscope.

**Apoptosis analysis.** Apoptosis was detected with flow cytometry analysis. Briefly, cells were collected by trypsinization at 48 h after treatment with IL-6, osimertinib, or both. Then, cells were washed three times with PBS, and resuspended at a density of $1 \times 10^6$ cells/mL. Next, cells were double-stained with PI for 10 min at ambient temperature in the dark using FITC Annexin V Apoptosis Detection Kit I (Sigma, Germany). Then, cells were analyzed using a flow cytometer (Beckman Coulter Navios, CA, USA).

**Western blot.** Cells harvested by scraping were washed twice with PBS and lysed for 30 min at 4 °C in RIPA buffer (Sigma-Aldrich, France). After centrifugation at $12000 \times g$ for 20 min at 4 °C, the protein content was determined by the BCA assay. Equal amounts of protein were submitted to gel electrophoresis for 2 h at 110 V, followed by transfer onto PVDF membranes (90 min, 200 mA) (Millipore, German). Membranes were blocked with 5% bovine serum albumin (BSA) for 1 h at room temperature and incubated overnight at 4 °C with primary antibodies. Next, the membranes were washed and then incubated with 0.02 μg/ml horseradish peroxidase (HRP)-conjugated goat anti-rabbit or anti-mouse IgG (Cell Signaling Technology, USA) for 1 h, and visualized with ChemiDoc Touch System (Bio-Rad, USA).

**Animal study.** Animal experiments were performed with the approval of the Committee on animal experimentation of the Army Medical University (Chongqing, China) and complied with all relevant ethical regulations. Briefly, $1 \times 10^6$ cells (PC-9GROR cells) were injected subcutaneously into the back, next to the left forelimb of 6-week-old female BALB/c A-nu mice (Laboratory Animal Center of Army Medical University). When tumors developed of ~30 mm³ within 5–7 days, mice were randomly assigned to groups of different treatments as indicated. Osimertinib (20 mg/L) or ibrutinib (25 mg/L) was given orally in drinking water (vehicle). The anti-IL6R antibody tocilizumab (100 mg/mice, i.p.) was administered twice a week. The tumor volume was calculated as (length × width²)/2, and measured twice a week. The animals were monitored for 4 weeks until euthanasia and sacrificed.

**Small molecule screening.** For high-throughput small molecule screenings, the *IL-6-GFP* PC-9 cells and *IL-6-GFP* PC-9GR cells were seeded in 96-well assay plates (Corning Incorporated, ME, USA; 3000 cells/well). The chemical library was purchased from Selleck (L2000, containing 510 compounds) and applied in a final concentration of 10 μM of each chemical in screening. A list of all compounds is provided in Supplementary Data 3. Two sets of plates were generated, one for GFP fluorescence read-out and the other for measurement of cell viability using the CCK-8 assay. Cells were treated with chemicals for 48 h and analyzed for GFP signals using a High-Content Screening System (Molecular Devices, USA). Cells treated with DMSO were used as controls. Two independent replicates were

performed for each compound screen and normalized data for GFP fluorescence and cell viability (defined as the fluorescence signal of GFP or OD value of each well divided by the control well) of each replicate is presented in Supplementary Data 3. The common compounds between the top 20 chemicals inhibiting IL-6 GFP fluorescence in each cell line were selected for further examination.

**Statistics and reproducibility**. Statistical analysis was performed by GraphPad Prism 8.0. Data are expressed as mean ± SEM. Student t-test was generally used to analyze the differences between two groups, but when the variances differed, the Mann–Whitney U test was used as indicated. A $p$-value <0.05 was considered to indicate statistical significance. We primarily assessed the causal relationship between baseline serum IL-6 levels and therapeutic efficacy, including PFS and overall survival (OS). PFS was defined as the duration from the date of initiation of EGFR-TKI treatment to the date of documented relapse. OS was defined as the duration from the date of initiation of EGFR-TKI treatment to the date of death. PFS and OS were assessed by the Kaplan–Meier method, and differences between KM curves were evaluated by the log-rank test. Details of statistical analysis and the number of biological replicates ($n$) can be found in each figure legend.

**Reporting summary**. Further information on research design is available in the Nature Research Reporting Summary linked to this article.

## Data availability

All data supporting the findings of this study are available within the article and its Supplementary Information files and from the corresponding authors upon reasonable request (Lead contact: heyong@tmmu.edu.cn). Source data are provided in Supplementary Data 4 and Supplementary Fig. 8. RNAseq dataset been deposited in GSA-Human with the HRA002482 accession number. The protein mass spectrum data has been deposited in the OMix (https://bigd.big.ac.cn/omix/) in NGDC with accession number OMIX868. Associated analyses are provided in Supplementary Data 1 and 2. The whole project has been registered in BioProject (https://bigd.big.ac.cn/bioproject/) in NGDC with accession number PRJCA007704.

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

## Acknowledgements

This work was supported by the National Natural Science foundation of China (81672287, 81972189, 81702291), the Science Foundation by Daping Hospital of Army Medical University (2019CXLCB011, 2019CXLCA003), and the Science Foundation for Outstanding Young People of the Army Medical University.

## Author contributions

Concept and design: L.L., H.C., and H.Y. Acquisition, analysis, or interpretation of data: L.L., L.Z.J., L.C.H., L.J., Z.K., L.C.Y., T.X., L.Z.L., and Z.Y.M. Drafting of the manuscript: L.L., L.Z.J., L.C.H., Z.Y.M., H.C., and H.Y. Critical revision of the manuscript for important intellectual content: All authors. Statistical analysis: L.L., L.Z.J., L.C.H., and L.J.H. Obtained funding: L.L. and H.Y. Administrative, technical, or material support: H.R., W.Y., F.M., and Z.Y. Supervision: L.L., H.C., and H.Y.

## Competing interests

The authors declare no competing interests.
