## [Peer Review File · Communications Biology]

Reviewers' comments:

Reviewer #1 (Remarks to the Author):

In this manuscript, Li and colleagues reported the IL-6-mediated resistance mechanism to Osimertinib, a third-generation EGFR tyrosine kinase inhibitor (EGFR-TKI). They used two lung cancer cell lines harboring EGFR mutations for in vitro and in vivo experiments. In addition, they presented some clinical data of IL-6 levels. They conclude laminin α 5/FAK signaling is responsible for IL-6-induced Osimertinib resistance. In general, the experiments were well performed. However, I have several concerns as addressed below.

1. As addressed by the authors, IL-6-mediated resistance to 1st and 2nd generation EGFR-TKIs have been previously reported, which makes the manuscript with limited novelty. I am wondering if there is a difference of mechanism between 1st/2nd generation EGFR-TKI and Osimertinib. If there is a rationale reason to distinguish between 1st/2nd EGFR-TKI and Osimertinib, the authors should describe it.

2. The authors showed clinical data regarding IL-6 in Figure 1C-G. Is IL-6 level information available in the routine clinical setting? The authors described in the Materials and Methods section that this study is a retrospective study. How could authors obtain and use serum samples for ELISA experiments (Figure 1E) retrospectively? The authors should explain this.

3. Do authors think blood IL-6 levels are derived from lung cancer cells? As IL-6 could be derived from multiple types of cells, high blood IL-6 levels could be derived from non-lung cancer cells. I think this is difficult for readers to understand. The authors should explain it.

4. In vitro/ in vivo experiments were only performed on two lung cancer cell lines, PC9 and H1975, which makes the story less robust.

5. The authors proposed ibrutinib to overcome IL-6 mediated resistance, however, the mechanism of action is not well evaluated. Is the effect of ibrutinib through inhibiting BTK? How does ibrutinib decrease the expression of IL-6? I think these inhibitors exert an off-target effect. The authors should perform the experiments that the IL-6 inhibition by ibrutinib is through BTK or something else. If it is through BTK, knockdown or knockout of BTK could cancel the inhibition. As the authors addressed, ibrutinib can also inhibit EGFR, the combination therapy with osimertinib and ibrutinib can just intensify the inhibition of EGFR signaling.

Reviewer #2 (Remarks to the Author):

The study by Li et al. in this submission has demonstrated that induction of IL-6/laminin α 5/FAK activation plays a critical role in mediating acquired resistance to osimertinib and blockage of this signaling axis with ibrutinib in part restores the sensitivity of EGFRm NSCLC cells/tumors to osimertinib. The experiments were in general well conducted with clear data that support their major conclusions. The finding with ibrutinib combined with osimertinib as a strategy for overcoming osimertinib acquired resistance is of immediate translational significance and can be tested in the clinic. The following are some recommendations for improving the manuscript.

1. One key finding in this study is the elevation of IL-6 induced by osimertinib and other EGR-TKIs. It will be nice to analyze the data presented in Fig. 1E to see whether patients with highly elevated IL-6 post TKI treatment do worse than those without or with limited increase of IL-6.

2. Are ELISA data presented in Fig. 1B from cell lysates or from medium (secreted)? If from cell lysates, can increase of secreted IL-6 in medium be detected in these resistant cell lines?

3. On page 6, line 145, "endogenous IL-6" should be "exogenous IL-6"? Moreover, the expression of the exogenous IL-6 in these stable cell lines should be shown.

4. In Fig. 1, Ki67 proliferation assay was used to evaluate the protective effects of IL-6. Given that osimertinib primarily induces apoptosis in EGFRm NSCLC cell lines. It is better to measure apoptosis such as with Annexin V/flow cytometry.

5. In Fig. 6A, the single agent activity of ibrutinib at 0.5 μ M on the survival of the tested cell lines

should be shown or described.

6. ELISA detection of IL-6 should be included in the Materials and Methods

7. On page 16, line 495, the drug dosage "mg/L" should be "mg/kg"?

Point-by-point response (Reviewers' comments in bold italics)

Reviewers' comments:

Reviewer #1 (Remarks to the Author):

In this manuscript, Li and colleagues reported the IL-6-mediated resistance mechanism to Osimertinib, a third-generation EGFR tyrosine kinase inhibitor (EGFR-TKI). They used two lung cancer cell lines harboring EGFR mutations for in vitro and in vivo experiments. In addition, they presented some clinical data of IL-6 levels. They conclude laminin α 5/FAK signaling is responsible for IL-6-induced Osimertinib resistance. In general, the experiments were well performed. However, I have several concerns as addressed below.

1. As addressed by the authors, IL-6-mediated resistance to 1st and 2nd generation EGFR-TKIs have been previously reported, which makes the manuscript with limited novelty. I am wondering if there is a difference of mechanism between 1st/2nd generation EGFR-TKI and Osimertinib. If there is a rationale reason to distinguish between 1st/2nd EGFR-TKI and Osimertinib, the authors should describe it.

Response: Thanks for the question. The drug resistance mechanisms of osimertinib are different from those of 1st and 2nd generation EGFR-TKIs. For example, EGFR T790M mutation is the most common resistance mechanism to 1st and 2nd generation EGFR-TKIs, while patients with this mutation are sensitive to osimertinib. The mechanisms of osimertinib resistance are more complex, possibly due to tumor evolution under the pressure of drug exposure. One recent study identified a total of 13 recurrent mutations which were only observed in osimertinib-resistant group, but not in 1st generation TKI group [1]. In the current study, we also found that the mechanisms of IL-6-mediated resistance to osimertinib are different from IL-6-mediated resistance to 1st and 2nd generation EGFR-TKIs. We and others have previously reported that IL-6 could induce resistance to 1st generation EGFR-TKI through activation of STAT3, and inhibition of STAT3 could reverse TKI resistance [2, 3]. In the current study, we found that inhibition of STAT3 failed to reverse osimertinib resistance, while siRNA-mediated knockdown of PTK2 enhanced osimertinib sensitivity. Taken together, these results, together with recent publications, suggest that there is a difference of mechanism between 1st/2nd generation EGFR-TKI and osimertinib. We have added the following sentences into the Results section: "Previously, it was reported that tocilizumab could overcome IL-6-induced resistance to erlotinib, a 1st generation EGFR-TKI. Also, we and others have reported that inhibition of STAT3 could reverse drug resistance to 1st generation EGFR-TKIs. These data suggest that there is a difference of mechanism between 1st/2nd generation EGFR-TKI and osimertinib. IL-6/IL-6R blockade or STAT3 inhibition alone was unable to reverse osimertinib resistance." (Pages 6-7, Lines 167-172)

References:

1. Lin L, Lu Q, Cao R, Ou Q, Ma Y, Bao H, Wu X, Shao Y, Wang Z, Shen B: Acquired rare recurrent EGFR mutations as mechanisms of resistance to Osimertinib in lung cancer and in silico structural modelling. *Am J Cancer Res* 2020, 10(11):4005-4015.
2. Li L, Han R, Xiao H, Lin C, Wang Y, Liu H, Li K, Chen H, Sun F, Yang Z et al: Metformin sensitizes EGFR-TKI-resistant human lung cancer cells in vitro and in vivo through inhibition of IL-6 signaling and EMT reversal. *Clin Cancer Res* 2014, 20(10):2714-2726.
3. Zheng Q, Dong H, Mo J, Zhang Y, Huang J, Ouyang S, Shi S, Zhu K, Qu X, Hu W et al: A novel STAT3 inhibitor W2014-S regresses human non-small cell lung cancer xenografts and sensitizes EGFR-TKI acquired resistance. *Theranostics* 2021, 11(2):824-840.

2. The authors showed clinical data regarding IL-6 in Figure 1C-G. Is IL-6 level information available in the routine clinical setting? The authors described in the Materials and Methods section that this study is a retrospective study. How could authors obtain and use serum samples for ELISA experiments (Figure 1E) retrospectively? The authors should explain this.

Response: Thanks for the question. It is important to clarify this issue. We confirm that this is a retrospective study. Serum IL-6 level was retrieved from medical records, which had been measured by Enzyme-linked immunosorbent assay (ELISA) at Department of Clinical Laboratory, Daping Hospital in clinical tests. Indeed, not every patient received this test. For example, as shown in the study flow chart of Fig.S1, of 93 patients with sequential osimertinib treatment, only 43 patients had pre-osimertinib IL-6 data. We have modified the sentence "Serum IL-6 level was retrieved from medical records, which had been measured by Enzyme-linked immunosorbent assay (ELISA) at Department of Clinical Laboratory, Daping Hospital in clinical tests." in the Methods section (Page 13, Lines 385-387).

3. Do authors think blood IL-6 levels are derived from lung cancer cells? As IL-6 could be derived from multiple types of cells, high blood IL-6 levels could be derived from non-lung cancer cells. I think this is difficult for readers to understand. The authors should explain it.

Response: Thanks for the question. We fully agree with the reviewer that blood IL-6 could be derived from multiple types of cells, including B cells, macrophages, and fibroblasts. On one hand, we have performed more experiments and proved that higher levels of IL-6 were found in culture medium as well as cell lysates of osimertinib-resistant cell lines by ELISA (updated Figure 1B). On the other hand, we added the following sentences into the Discussion section: "However, one should bear in mind that blood IL-6 could be derived from multiple types of cells other than cancer cells, such as B cells, macrophages, and fibroblasts [1-3]. Thus, a comprehensive analysis is needed to correlate IL-6 increase with osimertinib resistance." (Page11, Lines 310-313)

References:

1. Barr TA, Shen P, Brown S, et al. B cell depletion therapy ameliorates autoimmune disease through ablation of IL-6-producing B cells. *J Exp Med*. 2012;209(5):1001-1010.

2. Jung BG, Wang X, Yi N, Ma J, Turner J, Samten B. Early Secreted Antigenic Target of 6-kDa of Mycobacterium tuberculosis Stimulates IL-6 Production by Macrophages through Activation of STAT3. *Sci Rep.* 2017;7:40984.
3. Shintani Y, Fujiwara A, Kimura T, et al. IL-6 Secreted from Cancer-Associated Fibroblasts Mediates Chemoresistance in NSCLC by Increasing Epithelial-Mesenchymal Transition Signaling. *J Thorac Oncol.* 2016;11(9):1482-1492.

4. *In vitro/ in vivo experiments were only performed on two lung cancer cell lines, PC9 and H1975, which makes the story less robust.*

Thanks a lot for this nice suggestion. We do agree that introducing more cell lines will add value to the current study. We used a new cell line, H3255 cells, which harbor EGFR L858R mutation. We performed CCK8 cell viability assay, Ki67 incorporation assay, and flow cytometry analysis on H3255 cells to study the effect of IL-6 treatment on osimertinib efficacy. The results were in consistent with those in other osimertinib-sensitive cells. Besides, we performed Western Blot analysis and found that IL-6 treatment resulted in increased expression of Laminin α 5, phosphorylated STAT3, and phosphorylated FAK. Please find those results in Figure 2E, Figure 3C, Fig. S2, and Fig. S3.

Besides, we also performed more in vitro experiments on *IL-6-GFP-HCC827* cells. We found that siRNA knockdown of *LAMA5* or *PTK2* significantly enhanced cytotoxicity of osimertinib in *IL-6-GFP-HCC827* cells. Also, ibrutinib effectively enhanced osimertinib cytotoxicity and decreased the levels of phosphorylated STAT3, phosphorylated FAK, and Laminin α 5 in *IL-6-GFP-HCC827* cells. Those results were in consistent with those in other cell lines with IL-6 overexpression. We have added those results into Fig. S6.

5. *The authors proposed ibrutinib to overcome IL-6 mediated resistance, however, the mechanism of action is not well evaluated. Is the effect of ibrutinib through inhibiting BTK? How does ibrutinib decrease the expression of IL-6? I think these inhibitors exert an off-target effect. The authors should perform the experiments that the IL-6 inhibition by ibrutinib is through BTK or something else. If it is through BTK, knockdown or knockout of BTK could cancel the inhibition. As the authors addressed, ibrutinib can also inhibit EGFR, the combination therapy with osimertinib and ibrutinib can just intensify the inhibition of EGFR signaling.*

We thank the reviewer for this important question. As previously reported, BTK played a role in ibrutinib-mediated inhibition of IL-6 [1]. Ibrutinib could inhibit BTK activity and decrease IL-6 levels. However, in BTK^{Cys481Ser} cells ibrutinib failed to decrease either BTK activity or IL-6 levels. These results suggest that ibrutinib inhibits IL-6 levels through inhibition of BTK [1]. We therefore performed more experiments to study the role of BTK in ibrutinib's activity on osimertinib sensitivity and IL-6 levels. SiRNA knockdown of *BTK* in PC-9GROR cells or *IL-6-GFP-PC-9GR* cells significantly increased osimertinib sensitivity, and decreased IL-6 levels (Fig. S7). Interestingly, after siRNA-mediated knockdown of *BTK*, ibrutinib failed to further significantly increase osimertinib sensitivity or decrease IL-6 levels. These results suggest that ibrutinib inhibited IL-6 levels at least in a large part by

inhibiting BTK activity. We have added those results into the Results section (Page 10, Lines 281-285).

Refs:

1. Chen JG, Liu X, Munshi M, Xu L, Tsakmaklis N, Demos MG, Kofides A, Guerrero ML, Chan GG, Patterson CJ *et al*: BTK(Cys481Ser) drives ibrutinib resistance via ERK1/2 and protects BTK(wild-type) MYD88-mutated cells by a paracrine mechanism. *Blood* 2018, 131(18):2047-2059.

Reviewer #2 (Remarks to the Author):

The study by Li et al. in this submission has demonstrated that induction of IL-6/laminin α 5/FAK activation plays a critical role in mediating acquired resistance to osimertinib and blockage of this signaling axis with ibrutinib in part restores the sensitivity of EGFRm NSCLC cells/tumors to osimertinib. The experiments were in general well conducted with clear data that support their major conclusions. The finding with ibrutinib combined with osimertinib as a strategy for overcoming osimertinib acquired resistance is of immediate translational significance and can be tested in the clinic. The following are some recommendations for improving the manuscript.

Response: Thanks for the time and patience from the reviewer on the manuscript. We do agree with the reviewer about the following comments of this study. Please find our point-by-point response below.

1. One key finding in this study is the elevation of IL-6 induced by osimertinib and other EGR-TKIs. It will be nice to analyze the data presented in Fig. 1E to see whether patients with highly elevated IL-6 post TKI treatment do worse than those without or with limited increase of IL-6.

Response: Thanks for the nice suggestion. We have provided additional analysis, which was shown in Fig. S1B and C. Upon gefitinib resistance, patients with highly elevated IL-6 levels (value \geq 7mg/L upon resistance and higher than that of baseline) had a significantly shorter OS than those with decreased or slightly increased IL-6 levels (value < 7mg/L upon resistance), as shown in Fig. S1B. Upon osimertinib resistance, patients with highly elevated IL-6 levels had a non-significant shorter OS, possibly due to limited patient numbers (Fig. S1C). We have added those sentences into the Results section (Page 5, Lines 130-136).

2. Are ELISA data presented in Fig. 1B from cell lysates or from medium (secreted)? If from cell lysates, can increase of secreted IL-6 in medium be detected in these resistant cell lines?

Response: Thanks for the question. In the previous version of the manuscript, the ELISA data were from culture medium. According to this suggestion, we have measured IL-6 levels from cell lysates of those cell lines. Results were consistent with those from the culture medium. As shown in the updated Figure 1B, higher levels of IL-6 were found in cell lysates of osimertinib-resistant cell lines as compared with the parental cells. We have

added this result into the first paragraph of the Results section: higher levels of IL-6 were found in culture medium as well as cell lysates of osimertinib-resistant cell lines by ELISA (Page 5, Lines 111-112).

3. On page 6, line 145, “endogenous IL-6” should be “exogenous IL-6”? Moreover, the expression of the exogenous IL-6 in these stable cell lines should be shown.

Response: Thanks for the question. To avoid confusion, we have changed the word “endogenous” to “exogenous”. Besides, we have measured the levels of IL-6 in these stable cell lines. As shown in Fig.S2D, significantly higher levels of IL-6 were found in culture medium of cells with expression of the exogenous IL-6.

4. In Fig. 1, Ki67 proliferation assay was used to evaluate the protective effects of IL-6. Given that osimertinib primarily induces apoptosis in EGFRm NSCLC cell lines. It is better to measure apoptosis such as with Annexin V/flow cytometry.

Response: Thanks for the nice suggestion. We have performed flow cytometry analysis of Annexin V and PI to detect apoptosis in PC-9 cells and H3255 cells, treated with osimertinib, IL-6 or both. As shown in Fig. S3, osimertinib treatment induced apoptosis in both cell lines. However, in the presence of IL-6, osimertinib failed to induce apoptosis of these cells. These results indicate that IL-6 protected those cancer cells from the pro-apoptotic effect of osimertinib. We have added those results into the Results section (Page 6, Lines 154-158).

5. In Fig. 6A, the single agent activity of ibrutinib at 0.5 μM on the survival of the tested cell lines should be shown or described.

Response: Thanks for the question. The single agent activity of ibrutinib at 0.5 μM showed a slight decrease of cell viability in PC-9OR3 cells, PC-9GROR3 cells, *IL-6-GFP-PC-9* cells, *IL-6-GFP-PC-9GR* cells, *IL-6-GFP-HCC827* cells, H1975-OR6 cells, as well as H1975-OR7 cells. We have added those results into Page 10, Line 274.

6. ELISA detection of IL-6 should be included in the Materials and Methods

Response: Thanks for the question. We have added the ELISA detection of IL-6 into the Materials and Methods section (Page 14, Lines 410-413).

7. On page 16, line 495, the drug dosage “mg/L” should be “mg/kg”?

Response: Thanks for the question. We confirm that the drug dosage is “mg/L”, as used in previous studies [1]. Also, as previously reported by Iliopoulos and colleagues, in animal studies, 1 mg/mL drug corresponds to 75 mg/kg [2]. So, 20mg/L osimertinib corresponds to 2.5mg/kg. The well-established Reagan-Shaw method [3] shows the human equivalent dose (mg/kg) = animal dose (mg/kg) x animal Km/human Km, where Km values are based on body surface area. For a 60 kg human adult, Km is 37, whereas for a 20g mouse, it is 3. Thus, the human equivalent of the murine dose of 2.5 mg/kg is 12 mg in an adult of 60 kg, which is much less than the osimertinib dose of 80 mg/day recommended by the Food and Drug Administration. Thus, the in vivo dose of osimertinib in this study is within a therapeutic range in humans.

References:

1. Li L, Wang Y, Jiao L, Lin C, Lu C, Zhang K, Hu C, Ye J, Zhang D, Wu H et al: Protective autophagy decreases osimertinib cytotoxicity through regulation of stem cell-like properties in lung cancer. *Cancer Lett* 2019, 452:191-202.
2. Iliopoulos, D., H.A. Hirsch, and K. Struhl, Metformin decreases the dose of chemotherapy for prolonging tumor remission in mouse xenografts involving multiple cancer cell types. *Cancer Res*, 2011. 71(9): p. 3196-201.
3. Reagan-Shaw, S., M. Nihal, and N. Ahmad, Dose translation from animal to human studies revisited. *FASEB J*, 2008. 22(3): p. 659-61.

REVIEWERS' COMMENTS:

Reviewer #1 (Remarks to the Author):

I think the authors appropriately responded to my concerns.

Point-by-point response (Reviewers' comments in bold italics)

Reviewers' comments:

Reviewer #1 (Remarks to the Author):

I think the authors appropriately responded to my concerns.,

Response: We thank the Reviewer again for the valuable comments to improve our manuscript.